# Neural Isoforms of Agrin Are Generated by Reduced PTBP1−RNA Interaction Network Spanning the Neuron−Specific Splicing Regions in *AGRN*

**DOI:** 10.3390/ijms24087420

**Published:** 2023-04-18

**Authors:** Samira Bushra, Ying-Ni Lin, Atefeh Joudaki, Mikako Ito, Bisei Ohkawara, Kinji Ohno, Akio Masuda

**Affiliations:** Division of Neurogenetics, Center for Neurological Diseases and Cancer, Nagoya University Graduate School of Medicine, Nagoya 466-8550, Aichi, Japan

**Keywords:** agrin, alternative splicing, PTBP1, acetylcholine receptor clustering

## Abstract

Agrin is a heparan sulfate proteoglycan essential for the clustering of acetylcholine receptors at the neuromuscular junction. Neuron−specific isoforms of agrin are generated by alternative inclusion of three exons, called Y, Z8, and Z11 exons, although their processing mechanisms remain elusive. We found, by inspection of splicing *cis*−elements into the human *AGRN* gene, that binding sites for polypyrimidine tract binding protein 1 (PTBP1) were extensively enriched around Y and Z exons. *PTBP1*−silencing enhanced the coordinated inclusion of Y and Z exons in human SH−SY5Y neuronal cells, even though three constitutive exons are flanked by these alternative exons. Deletion analysis using minigenes identified five PTBP1−binding sites with remarkable splicing repression activities around Y and Z exons. Furthermore, artificial tethering experiments indicated that binding of a single PTBP1 molecule to any of these sites represses nearby Y or Z exons as well as the other distal exons. The RRM4 domain of PTBP1, which is required for looping out a target RNA segment, was likely to play a crucial role in the repression. Neuronal differentiation downregulates PTBP1 expression and promotes the coordinated inclusion of Y and Z exons. We propose that the reduction in the PTPB1−RNA network spanning these alternative exons is essential for the generation of the neuron−specific agrin isoforms.

## 1. Introduction

Agrin is an extracellular matrix protein, expressed as multiple isoforms in diverse tissues, including skeletal muscle, heart, lung, kidney, brain, blood vessels, immune cells, and bone [1]. Agrin drives various cellular processes, such as synaptic development, skeletal growth, and T cell−mediated immunity [2]. Among them, agrin is best characterized as an inducer of acetylcholine receptor (AChR) clusters required for the formation of neuromuscular junctions (NMJ) [3]. Neuron−specific agrin isoforms are generated through alternative splicing of *AGRN* pre−mRNA [4], and are exclusively capable of inducing AChR clustering. Pathogenic variants of *AGRN* compromise AChR clustering and cause congenital myasthenic syndrome (CMS) [5,6]. In addition, mis−splicing and reduced expression of *Agrn* were observed in model mice for spinal muscular atrophy (SMA) [7]. Restoration of the neuron−specific agrin isoform ameliorated muscle atrophy and motor impairment and prolonged the survival of these mice [8,9].

Human and mouse *AGRN* are huge genes encompassing 36,018 bp and 32,251 bp with 39 and 38 exons, respectively. In 1993, three alternatively spliced regions (X, Y, and Z sites) were identified in the rat *Agrn* gene, harboring 39 exons [10]. The X site generates alternative 5′ ends of exon 20 in rat *Agrn*, but this site is not alternatively spliced in other species [11]. The Y site contains a 12−nt alternative cassette exon, currently referred to as Y exon, which encodes four amino acids that confer the interactions with heparin and dystroglycan [10]. Alternative inclusion and skipping of the Y exon generate the Y4 and Y0 isoforms, respectively (Figure 1A). The Z site contains two short alternative cassette exons, termed 24−nt Z8 and 33−nt Z11 exons [4,10]. Inclusion of Z8, Z11, and both Z8 and Z11 exons, generate the Z8, Z11, and Z19 isoforms, respectively, which are crucial for AChR clustering (Figure 1A) [4,10].

The neuron−specific inclusion of Y and Z exons has been identified in rat, chicken, mouse, and zebrafish [10,12,13,14,15]. Splicing of Y, Z8, and Z11 exons is coordinately regulated in rat, resulting in abundant expression of Y4Z8 and Y4Z19 isoforms and minimal expression of other isoforms (Y4Z11 and Y0Z0/8/11/19) in neural tissues [10,16]. In contrast, the Y0Z0 isoform is widely expressed in non−neural tissues. The neuron−specific isoforms, Y4Z8 and Y4Z19, can induce AChR clustering quite efficiently (~50−fold more potent than the Y4Z11 isoform), whereas the non−neural Y0Z0 isoform is incapable of clustering AChRs [1,10,13,16]. Despite the essential role of the neural isoforms of agrin in AChR clustering, the splicing−regulating mechanism(s) to generate neuron−specific isoforms of agrin remain to be disclosed. In addition, the splicing patterns of Y and Z exons in human cells and tissues have not been investigated. A previous study identified an intronic splicing enhancer downstream of the Y exon in mouse, but its cognate splicing *trans*−factor remains unidentified [17]. Knockouts of splicing *trans*−factors, such as NOVA1/2 and SNRNP70, reduced the inclusion of Z exons in neuronal tissues, but their cognate splicing *cis*−elements in *Agrn* have not been demonstrated [12,15].

Polypyrimidine tract binding protein 1 (PTBP1) is a ubiquitously expressed RNA binding protein that recognizes CU−rich sequences in pre−mRNAs to repress splicing of many alternative exons [18]. Several mechanisms have been proposed for the splicing−repressive activity of PTBP1, including antagonizing the function of the essential splicing factor U2AF65 [18], looping of RNA to inhibit spliceosome access [19], and multimeric binding of PTBP1 around a target exon to mask regulatory sequences required for splicing [20]. In neurons, the expression of PTBP1 is reduced during neuronal differentiation, while that of the neuronal paralog PTBP2 is gradually increased [21]. PTBP1 and PTBP2 repress splicing of overlapping but distinct sets of exons, and the switching from PTBP1 to PTBP2 promotes the expressions of neuron−specific mRNA isoforms by altering the alternative splicing fate of many exons [22,23].

Here we show, by analysis of various human tissues and cells, that *AGRN* Y and Z exons are coordinately included in differentiated human neuronal cells. Multiple PTBP1−binding sites are widely distributed around these exons, and knockdown experiments revealed that PTBP1 but not PTBP2 is essential for repressing the coordinated splicing. Minigene analysis revealed primary and compensatory roles of these PTBP1−binding sites in repressing nearby Y or Z exons. Furthermore, these sites also mediate PTBP1−dependent repression of the other distant exon(s) despite the presence of three constitutive exons between Y and Z exons (Figure 1A). Multiple PTBP1 bindings around Y/Z exons form a complex network of repressive splicing mechanisms, and downregulation of PTBP1 is essential for the production of neural agrin isoforms.

## 2. Results

### 2.1. Alternative Splicing of AGRN Y and Z Exons in Human Cells and Tissues

We first analyzed the splicing patterns of Y and Z exons of *AGRN* in various human tissues using RT−PCR. As has been previously reported in mice and rats [10,13], Y and Z exons were included in the human brain (Figure 1B, upper two panels). Similarly, RNA−seq analysis of 54 human tissues by the Genotype−Tissue Expression (GTEx) Consortium [24] also showed specific inclusion of Y and Z exons in the brain (Appendix A). Considering the essential roles of NMJs in cholinergic signaling [10,11], we next asked whether the inclusion of Y and Z exons is predominant in cholinergic neurons. We observed that Y and Z exons are preferentially included in brain tissue regardless of the expression levels of the cholinergic markers *CHAT* and *VACHT* [25] (Appendix A). Although sex differences have been observed in the cholinergic signaling system [26], the inclusion levels of Y and Z exons are similar between male and female samples in all the analyzed tissues (Appendix A). These results suggest that the expression of the neural agrin isoforms is not restricted to cholinergic neurons.

Further, RT−PCR analysis revealed expression of the Z8 and Z19 but not Z11 isoform in the human brain (Figure 1B), indicating coordinated splicing of the Z11 exon with the Z8 exon, as was observed in rat [10]. We further analyzed coordinated splicing of Y and Z exons using nested RT−PCR. We first amplified transcripts including and lacking exon Y using isoform−specific primers (Y+ and Y− transcripts, respectively), and then amplified a segment spanning Z exons (Appendix A). We observed that Z exons were included in the Y+ transcripts, but not in the Y− transcripts in the human brain (Figure 1B, lower three panels). In the other tissues, neither Y nor Z exons were included and only the Y0Z0 isoform was observed.

Next, we induced neuronal differentiation of the SH−SY5Y human neuroblastoma cell line by retinoic acid supplementation. We confirmed the upregulation of the neuronal differentiation markers *NSE* and *GAP43* (Appendix A). RT−PCR analysis showed that the inclusion of Y and Z exons was gradually increased with neuronal differentiation (Figure 1C–E). The inclusion of Z8 and Z19 but not Z11 was also increased during differentiation, especially in Y+ transcripts (Figure 1C,F,G).

These results indicate that the Y, Z8, and Z11 exons are coordinately included to produce the Y4Z8 and Y4Z19 isoforms in differentiated human neuronal cells and in the human brain.

### 2.2. Identification of PTBP1 as a Repressor of Alternative Splicing of Y and Z Exons

To identify splicing *cis*−elements involved in the alternative splicing of *AGRN*, we analyzed distribution of all possible di−nucleotide motifs from exons 30 to 37 in the human *AGRN* gene, whereby exons 31, 35, and 36 are Y4, Z8, and Z11 exons, respectively. We observed the enrichment of the CU/UC motif around Y, Z8, and Z11 exons, especially upstream of the exons (Figure 2A, upper panel). Although the GU/UG motif is enriched in the deep intron between the Z8 and Z11 exons, this region is not well conserved in other species (Figure 2A, lower panel), suggesting that the GU/UG motif is dispensable. CU−rich sequences are well known as binding sites for the splicing repressor PTBP1 [27]. Consistently, eCLIP analysis of PTBP1 by the Encyclopedia of DNA Elements (ENCODE) Consortium [28,29,30] showed the accumulation of PTBP1 bindings around Z exons in K562 cells (Appendix A). Moreover, SpliceAid 2, a database for target sequences of RBPs [31], predicted the presence of multiple PTBP1−binding sites around Y and Z exons (Appendix A).

To investigate the involvement of PTBP1 in the splicing of Y and Z exons, we knocked down PTBP1 using specific siRNA in undifferentiated SH−SY5Y cells. Immunoblotting showed that siRNA against *PTBP1* efficiently downregulated the expression of PTBP1 protein, while its neuronal paralog, PTBP2, was upregulated (Figure 2B), as previously reported [32]. Double knockdown of PTBP1 and PTBP2 (PTBP1 + 2 silencing) efficiently silenced both proteins. As expected, knockdown of PTBP1 increased the inclusion of Y and Z exons (lane 2 in Figure 2B–D). Knockdown of PTBP1 and of PTBP1 + 2 significantly enhanced the inclusion of the Y exon (lanes 2 and 4 in Figure 2B,C), and of Z exons to a lesser extent (lanes 2 and 4 in Figure 2B,D). In contrast, PTBP2 silencing had no essential effect on the inclusion of Y and Z exons (lane 3 in Figure 2B–D). In addition, silencing of PTBP1 and of PTBP1 + 2 enhanced the inclusion of Z exons in Y+ transcripts but not in Y– transcripts (lanes 2 and 4 in Figure 2B,E,F). These results indicate a repressive effect of PTBP1 on the splicing of Y and Z exons. Consistently, PTBP1 expression was reduced in differentiating SH−SY5Y cells (Appendix A), while the inclusion of Y and Z exons was enhanced (Figure 1C). The negative relationship between *PTBP1* expression and the inclusion level of Y and Z exons was also observed in the GTEx RNA−seq data of 54 human tissues (*R* = −0.66) (Appendix A).

In addition to PTBP1, SpliceAid 2 showed presence of binding sites for CELF2, YB−1, HNRNP H/F, and SRSF2 around Y and Z exons (Appendix A). Individual overexpression of CELF2 and SRSF2 or knockdown of YB−1 and HNRNP H/F did not alter the splicing of either Y or Z exons (Appendix A).

### 2.3. Specific Roles of PTBP1−Binding Sites around Y and Z Exons

To understand how PTBP1 represses the splicing of Y and Z exons, we next constructed a minigene harboring exons 30 to 37 of human *AGRN* in a mammalian expression vector (Appendix A). Similar to endogenous *AGRN* mRNA, silencing of PTBP1 and of PTBP1 + 2, but not of PTBP2, enhanced the splicing of both Y (Appendix A) and Z (Appendix A) exons in Y+ transcripts of the *AGRN* minigene.

SpliceAid 2 predicted five possible pyrimidine−rich binding sites for PTBP1 (P−Y1 to P−Y5 sites) around the Y exon (Appendix A). Notably, the P−Y1 to P−Y4 sites were located within a polypyrimidine (PY) tract (~150 bp long), which is essential for 3′ splice site recognition. To identify specific PTBP1−binding site(s) that mediate the splicing repression of the Y exon, we individually mutated these five sites in the *AGRN* minigene (Figure 3A). RT−PCR analysis showed that disruption of the P−Y2 site (P−Y2 MG) prominently enhanced the inclusion of the Y exon (Figure 3A,B). Disruption of the P−Y1 and P−Y4 sites mildly enhanced it, whereas disruption of the P−Y3 and P−Y5 sites had minimal effects. Interestingly, disruption of the P−Y1 site (P−Y1 MG) remotely enhanced the inclusion of Z exons, (Figure 3A,C), although the P−Y1 site is separated by multiple constitutive exons and introns from the Z exons (Figure 1A).

We next tethered PTBP1 to the P−Y2 and P−Y1 sites using the bacteriophage MS2 coat protein to further confirm the importance of these sites in mediating PTBP1−dependent repression of the splicing. We first replaced the P−Y2 site with the MS2 hairpin−loop sequence in the wild−type *AGRN* minigene to generate P−Y2 MS2hp MG (Figure 3D, top). Then, the minigene was introduced into SH−SY5Y cells together with a plasmid expressing PTBP1 fused with MS2 coat protein (PTBP1−MS2) to artificially bind PTBP1 to the P−Y2 site. As controls, we expressed SRSF1 fused with MS2 coat protein (SRSF1−MS2) or PTBP1/2 without MS2 coat protein (PTBP1 or PTBP2). Consistent with the mutation analysis (Figure 3A), tethering of PTBP1−MS2 to the P−Y2 site prominently induced the skipping of the Y exon but not the Z exons (Figure 3D–F, lane 4). PTBP2−MS2 also induced the skipping of the Y exon (Appendix A), although the effect was much less than that of PTBP1−MS2. Similar to PTBP1, PTBP2 did not induce skipping of the Z exons (Appendix A). In summary, our analysis revealed that the P−Y2 site preferentially mediates PTBP1−dependent repression of the Y exon.

We next tethered PTBP1 to the P−Y1 site (P−Y1 MS2hp MG). We noticed that both Y and Z exons were mostly skipped in P−Y1 MS2hp MG, even in the control experiments without tethering PTBP1 (Appendix A), probably due to PTBP1 binding to the other PTBP1−binding sites. We thus knocked down *PTBP1* and *PTBP2* to minimize the effects of PTBP1−binding sites other than P−Y1. In addition, we disrupted the P−Y2 site in the minigene (P−Y2mut/P−Y1 MS2hp−MG) to completely eliminate its effect, since the P−Y2 site has a primary role in the repression of the Y exon as stated above. As expected, the combination of P−Y2 mutagenesis and PTPB1/2 knockdown resulted in a marked increase in the inclusion of the Y and Z exons (Figure 3G–I, lane 1), and PTBP1−MS2 significantly induced skipping of the Y exon (Figure 3GH, lane 4) and the Z exons (Figure 3G,I, lane 4) in P−Y2mut/P−Y1 MS2hp−MG, which was consistent with the mutagenesis experiments (Figure 3A–C, lane 2). These results suggest the involvement of the P−Y1 site in the repression of the proximal Y exon and the distal Z8 and Z11 exons.

Unmasking the effects of PTBP1−tethering on the P−Y1 site by the disruption of the P−Y2 site along with knockdown of *PTBP1/PTBP2* suggests that PTBP1 binding sites redundantly repress Y and Z exons and they might have compensatory roles for each other. Thus, we next examined the effects of the P−Y2 site on Z exons. Similar to the tethering experiment for P−Y1, we disrupted the P−Y1 site in the minigene and tethered PTBP1 at the P−Y2 site while both *PTBP1* and *PTBP2* were knocked down. RT−PCR analysis revealed that the P−Y2 site also contributed to the repression of Z exons (Appendix A), which was masked by PTBP1 binding to the other sites in previous experiments (Figure 3D,F, lane 4).

We also analyzed the effect of PTBP1 tethering around Z exons on its splicing. SpliceAid−2 predicted six PTBP1 binding sites (P−Z1 to P−Z6) around Z8 and Z11 (Appendix A). We observed that individual disruptions of the P−Z1, P−Z3, P−Z4, and P−Z5 sites in the minigene enhanced the inclusion of Z exons (Figure 4A,C). Consistently, tethering of PTBP1 at the P−Z1, P−Z3, and P−Z5 sites inhibited the inclusion of Z exons in cells treated with siPTBP1 + 2 (P−Z1, Figure 4D,F; P−Z3, Appendix A; and P−Z5, Figure 4G,I). The tethering also repressed the inclusion of the remote Y exon (P−Z1, Figure 4D,E; P−Z3, Appendix A; and P−Z5, Figure 4G,H).

Taken together, our mutagenesis and tethering analyses demonstrate that PTBP1 represses splicing of both Y and Z exons through multiple binding sites around them in *AGRN* pre−mRNA. These sites have redundant and compensatory activities for the splicing repression of Y and Z exons, even though these exons are separated by three constitutive exons.

### 2.4. The Essential Role of RRM4 in PTBP1 for the Repression of Both Y and Z Exons

Finally, we investigated the mechanism of how PTBP1 binding to P−Y or P−Z sites represses both Y and Z exons, which are located distally from each other. Since one molecule of MS2 coat protein binds to one MS2−hairpin RNA, our tethering experiments suggest that a single PTBP1 molecule, binding to one of the PTBP1 binding sites, promotes the repression of both Y and Z exons in *AGRN* pre−mRNA. Therefore, we focused on the domains of PTBP1 which would be responsible for the repression (Figure 5A). Among the four RNA−recognition−motif (RRM) domains in PTBP1, RRM4 is essential for the splicing repression by PTBP1 through the looping out mechanism [19,20,33]. RRM3 and RRM4 have a fixed orientation to each other due to the rigid linker between them, which enables RRM4 to bind to a distal binding site and to form a hairpin loop of the bound RNA segment [33]. Indeed, PTBP1 represses splicing of a cassette exon by looping out the exon or the associated branch site in an RRM4−dependent manner [19,34]. To analyze the involvement of the looping−out mechanism in the repression of both Y and Z exons, we made a construct (PTBP1−RRM4del−MS2) to express an MS2−fused truncated PTBP1 protein lacking RRM4 (Figure 5A). We tethered PTBP1 or PTBP1−RRM4del at the P−Y1, P−Y2, P−Z1, P−Z3, and P−Z5 sites, where PTBP1 binds to repress both Y and Z exons as shown above. Our analysis revealed that PTBP1−RRM4del no longer represses the inclusion of either Y or Z exons through any of the analyzed PTBP−1 binding sites (P−Y1, Figure 5B–D; P−Y2, Appendix A; P−Z1, S10D–F; P−Z3, S10G–I; and P−Z5, Figure 5E–G). These results suggest the presence of RRM4−mediated looping−out mechanism in the repression of distal exons in *AGRN.*

## 3. Discussion

Here, we showed that Y and Z exons of *AGRN* are coordinately included in the human brain and differentiated neuronal cells. Pyrimidine−rich sequences are extensively enriched around these exons to recruit the splicing repressor PTBP1. Each of the pyrimidine−rich sites is capable of mediating PTBP1−dependent repression of both Y and Z exons to variable degrees, forming a complex network to repress the coordinated splicing of these exons (Figure 6A). RRM4 of PTBP1 is essential for the repression, suggesting that the looping−out mechanism is operational for the splicing repression. During neuronal differentiation, PTBP1 is downregulated, which subsequently enhances splicing of Y and Z exons to generate neuronal isoforms of agrin (Figure 6B).

The Y (12 nt), Z8 (24 nt), and Z11 (33 nt) exons are all small in size, although they encode amino acids having significant roles in shaping protein–protein interactions associated with neurogenesis, axon genesis, and synaptic functions [1,35,36]. Previous studies have shown that alternatively spliced short exons, so−called microexons, are highly conserved across species and are extensively included in mature neuronal cells [37,38]. Several RBPs, such as SRRM4, RBFOX, NOVA, and PTBP1, have been identified to regulate alternative splicing of microexons [37,38,39]. In particular, knocking out *Nova1/2* in mice was shown to disrupt the inclusion of Z exons [12]. NOVA is specifically expressed in neuronal tissues [12], in contrast to the ubiquitously expressed PTBP1 [20], although both RBPs often target the same exons for splicing [40]. These observations suggest that the neuron−specific inclusions of Y and Z exons may be accelerated by the expression of NOVA following the reduced expression of PTBP1 during neuronal differentiation. Additionally, our experiments showed that the repressive activity of PTBP2 for these exons is much lower than that of PTBP1 (Figure 3E and Appendix A). Although PTBP2 shares more than 70% amino acid similarity with PTBP1 [22,23,32], a protein–protein interaction study revealed that PTBP1 interacts with many splicing factors that PTBP2 does not interact with [41], which is likely to account for the distinct splicing activities between PTBP1 and PTBP2. Thus, substitution of PTBP2 for PTBP1 during neuronal differentiation (Appendix A) fine−tunes the splicing of Y and Z exons to produce neuron−specific agrin isoforms.

Several mechanisms have been proposed for the splicing repression by PTBP1 [18,19,20,27], including the competition with U2AF65 for binding to a PY tract, the multimeric binding of PTBP1 to mask regulatory sequences for splicing, and the looping out of an RNA segment to interfere with spliceosome access.

Our minigene analyses revealed primary roles of the P−Y2 and P−Z1 sites in the repression of nearby Y and Z8 exons, respectively (Figure 3B and Figure 4A). These sites are located in PY tracts for Y and Z8 exons (Appendix A). We observed, using in vitro experiments, that the recombinant proteins of PTBP1 and U2AF65 compete for the binding to the RNA probe (P−Y123) containing the P−Y2 site (Appendix A). Thus, PTBP1 binding to the P−Y2 site suppresses the splicing of the Y exon by competing against U2AF65. The repressive effect by competition is, however, limited to an exon immediately downstream of the target PY tract and does not affect a remote exon.

PTBP1 binding sites are often widely distributed in both the upstream and downstream intronic regions of target exons [27,42]. Similarly, multiple PTBP1−binding sites were identified around both Y and Z exons (Figure 6A). MS2−tethering analysis revealed that each PTBP1 binding site can mediate the PTBP1−dependent repression of both Y and Z exons through the looping−out mechanism. Thus, the multimeric PTBP1 binding around these exons forms redundant loops, making the splicing repression compensatory and robust. Interestingly, the Y and Z11 exons have PTBP1 binding sites in both flanking intronic regions, whereas Z8 has them only in the upstream region (Figure 6A). Therefore, the loop formation to repress splicing of Z8 exon should occur between P−Z1/2 upstream of the Z8 exon and the P−Z sites upstream and downstream of the Z11 exon. This may be the reason why the Z8 and Z11 exons are often coordinately included upon the reduction in PTBP1 expression (Figure 2B). A recent high−throughput sequencing analysis revealed that PTBP1 preferentially promotes RNA−loop formation around cassette exons in various genes [43]. These loops may also repress splicing of exons distal to the target cassette exons, as was observed in *AGRN*. As far as we know, this study provides the first evidence showing that PTBP1 represses the splicing of distal exons. We showed the requirement of RRM4 for the splicing repression, where the fixed orientation between RRM3 and RRM4 induces RNA−loop formation [19,33,34]. Similarly, the NMR analysis of hnRNP A1 revealed that hnRNP A1 loops out an RNA segment through its two tandem RRMs, which also have a fixed orientation [44]. Similar to PTBP1, hnRNP A1 may repress distal exons in addition to repressing cassette exons.

Agrin has been proposed as a potential therapeutic target for NMJ disorders due to its essential activity in the formation and maintenance of the NMJ [6,8,9]. Administration of NT−1654, a C−terminal fragment of the neuron−specific agrin isoform, promoted the clustering of AChRs at NMJs and ameliorated muscle atrophy in model mice of SMA [8], CMS [45], and myasthenia gravis [46]. Motor−neuron−specific transgenic expression of the neuron−specific agrin isoform also improved the muscle innervation and prolonged the survival of SMA model mice [9]. We here demonstrate that PTBP1 downregulation enhances the expression of the neuron−specific agrin isoform. Our analysis showed that nearly 50% of *AGRN* transcripts in neuronal tissues are still non−neural isoforms, lacking Y and Z exons (Appendix A), suggesting that enhancing splicing of Y and Z exons by targeting PTBP1 could be a promising therapeutic option for NMJ disorders. In addition to *PTBP1*−knockdown, masking the PTBP1 *cis*−elements by antisense oligonucleotide (ASO) would be a possible therapeutic modality, as used in the treatment of SMA against the *cis*−element for hnRNP A1 [47].

## 4. Materials and Methods

### 4.1. Cell Culture

SH−SY5Y human neuroblastoma cells were cultured in Dulbecco’s modified Eagle’s medium (DMEM, Gibco, Thermo Fisher Scientific, Grand Island, NY, USA) supplemented with 10% fetal bovine serum (FBS, Gibco, Thermo Fisher Scientific, Grand Island, NY, USA). To differentiate SH−SY5Y cells, the culture medium was replaced with DMEM containing 1% FBS and 10 μM retinoid acid.

### 4.2. Construction of Minigenes and Expression Vectors for Splicing Analysis

The *AGRN* minigene was constructed by amplifying exons 30 to 37 of human *AGRN* (genome coordinated from positions 1,050,763 to 1,053,794 of chromosome 1 according to GRCh38) using a proofreading DNA polymerase (PrimeSTAR, Takara, Kusatsu, Shiga, Japan). The forward primer specific for *AGRN* exon 30 contained an EcoRI restriction site followed by the Kozak consensus sequence (5′−CACCATG−3′) at the 5′ end. The reverse primer carried a stop codon and an NotI restriction site at the 5′ end. The amplified fragment was cloned into pcDNA3.1(+) vector at EcoRI and NotI sites. Artificial mutations and insertions were introduced into the *AGRN* minigene using a QuikChange Site−Directed Mutagenesis Kit (Agilent, Santa Clara, California, United States). Human *PTBP1* (accession number, BC013694), *PTBP2* (BC016582), *CELF2* (BC036391.1), *SRSF2* (BC001303.2) CDSs were purchased from Open Biosystems (Thermo Fisher Scientific, Huntsville, Alabama, United States) and were cloned into a p3xFLAG−CMV−14 mammalian expression vector. Primer sequences are listed in Appendix A.

### 4.3. Transfection of siRNAs and Expression Vectors

Cells were transfected with plasmids using FuGENE 6 (Promega, Madison, Wisconsin, United States), or with siRNAs using Lipofectamine RNAiMAX transfection reagent (Invitrogen, Thermo Fisher Scientific, Lithuania), according to the manufacturers’ instructions. siRNA sequences are shown in Appendix A.

### 4.4. RNA Extraction and RT−PCR

Total RNA was isolated at 48 h after transfection using RNeasy Mini Kit (Qiagen, Hilden, Germany) according to the manufacturer’s instructions, and then reverse−transcribed to make first−strand cDNA using an oligo−dT primer (Invitrogen, Thermo Fisher Scientific, Frederick, Maryland, United States) and reverse transcriptase (ReverTraAce, Toyobo, Osaka, Japan). PCR was performed with GoTaq polymerase (Promega, Madison, Wisconsin, United States ) using primers shown in Appendix A. RNA samples from human tissues were purchased from Clontech (Takara, Kusatsu, Shiga, Japan).

To analyze the coordinated splicing of Y and Z exons, nested RT−PCR was performed as indicated in Appendix A. Y+ (transcript harboring exon Y) and Y− (transcript lacking exon Y) were first independently amplified in a splicing isoform−specific manner using two distinct forward primers spanning exons 30 to 32 in combination with a reverse primer in exon 37. The forward primers contained and lacked the sequence of Y exon to specifically amplify Y+ and Y− transcripts, respectively. We confirmed these primer sets do not amplify undesired Y+ or Y− transcripts. Next, a second PCR was performed taking 1/10th of the 1st PCR−products as template with a forward primer in exon 34 and the same reverse primer, to examine specific inclusion of the Z exons in Y+ or Y− transcripts.

The intensities of PCR−amplified products were quantified with imageJ 1.53e (https://imagej.nih.gov/ij/index.html, accessed on 16 September 2020) to calculate %exon inclusion using the following equation:

%exon inclusion = [exon−included product/(exon−included product + exon−skipped product)] × 100

### 4.5. Immunoblotting

Cells were harvested using centrifugation at 2000× *g* for 5 min and suspended in lysis buffer (10 mM HEPES−NaOH (pH 7.8), 0.1 mM EDTA, 10 mM KCl, 1 mM DTT, 0.5 mM PMSF, and 0.1% NP−40) with protease inhibitors (1 μg/μL aprotinin, 1 μg/μL leupeptin, and 1 mM PMSF). Following sonication for 30 s with an ultrasonic disruptor (UR−20P, Tomy Seiko, Tokyo, Japan), the samples were centrifuged, and the supernatants were harvested as total cell lysates. SDS−PAGE and Western blotting were performed as previously described [48] with primary antibodies listed in Appendix A.

### 4.6. Tethered Function Assay of PTBP1 and PTBP2

To make an *AGRN* minigene carrying an MS2 hairpin loop at the P−Y1 (P−Y1 MS2hp−MG) or P−Y2 (P−Y2 MS2−hm−MG) site, the MS2 hairpin−loop sequence (5′−ACATGAGGATCACCCATGT−3′) was introduced at the respective site using a QuikChange Site−Directed Mutagenesis Kit (Agilent, Santa Clara, California, United States). Similarly, we made P−Y2mut/P−Y1 MS2hp−MG, P−Y1mut/P−Y2 MS2hp−MG, P−Z2mut/P−Z1 MS2hp−MG, P−Z4mut/P−Z3 MS2hp−MG, and P−Z6mut/P−Z5 MS2hp−MG, which have mutations at the P−Y2, P−Y1, P−Z2, P−Z4, and P−Z6 sites, respectively, along with the MS2 hairpin−loop sequence at the P−Y1, P−Y2, P−Z1, P−Z3, and P−Z5 sites, respectively. To make mammalian vectors expressing the PTBP1−MS2 and PTBP2−MS2 fusion proteins, the coding sequence of MS2 coat protein was amplified from a vector carrying the MS2−HNRNPH fusion cDNA [49], and was inserted into the 3′ ends of coding regions of PTBP1 and PTBP2 in p3xFLAG−CMV−14. PTBP1−RRM4del−MS2 construct was made by deleting the coding sequence of the RRM4 domain (from position 1463 to 1681 of BC013694 CDS) from PTBP1−MS2 using a QuikChange Site−Directed Mutagenesis Kit (Agilent, Santa Clara, California, United States). As a control, SRSF1 was fused with the MS2 coat protein (SRSF1−MS2) [50], since overexpression of SRSF1 had no effect on the splicing of Y exon and Z exons (Appendix A). Artificial tethering of PTBP1 and PTBP2 was performed by co−transfection of an *AGRN* minigene carrying an MS2 hairpin and an effector construct with or without the MS2 coat protein.

## Figures and Tables

**Figure 1 ijms-24-07420-f001:**
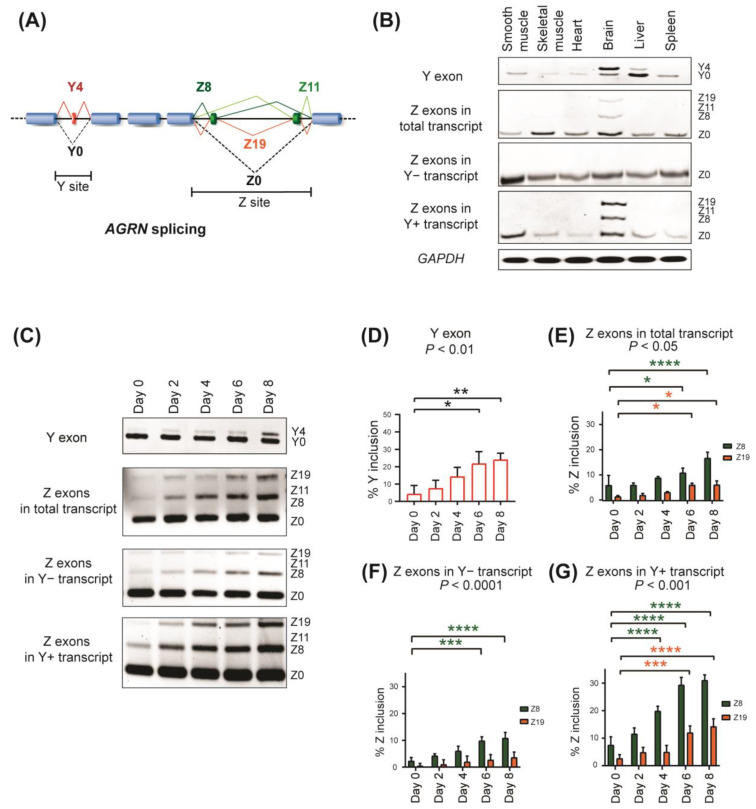
Alternative splicing of human *AGRN*. (**A**) Schematic showing alternative splicing patterns of *AGRN* at the Y and Z sites. (**B**) RT−PCR showing alternative splicing of Y and Z exons in various human tissues. (**C**) RT−PCR showing alternative splicing of endogenous Y and Z exons in SH−SY5Y cells in the course of neuronal differentiation. (**D**–**G**) Quantification of (**C**). Percent of Y exon inclusion (%Y inclusion) is shown in (**D**). Z exon inclusion (%Z inclusion) from total transcript, Y− transcript, and Y+ transcript is shown in (**E**–**G**), respectively. Mean and SD (*n* = 3 independent experiments) are indicated. *p*−values of one−way ANOVA (**D**) and two−way ANOVA (**E**–**G**) are indicated above the graphs. * *p* < 0.05, ** *p* < 0.01, *** *p* < 0.001, and **** *p* < 0.0001 using Tukey’s post hoc test compared to day 0 when applicable.

**Figure 2 ijms-24-07420-f002:**
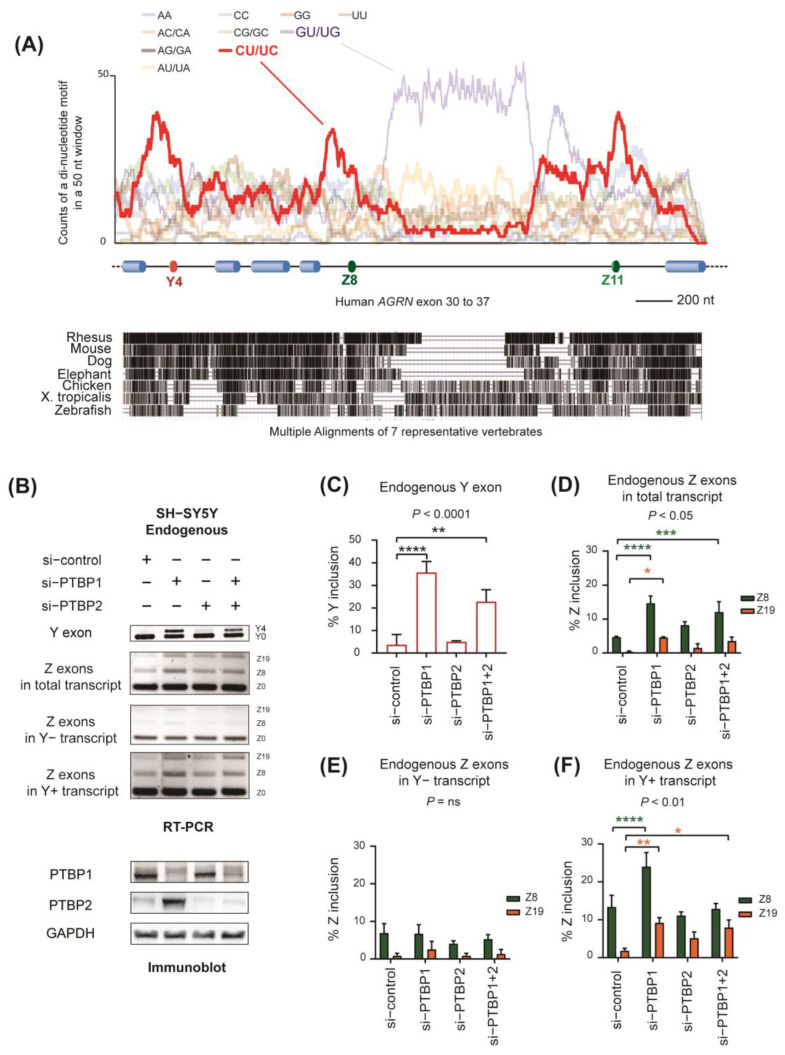
PTBP1 represses splicing of Y and Z exons. (**A**) (**Upper panel**) Enrichment of di−nucleotide motifs in human *AGRN* exons 30 to 37. The count of each di−nucleotide motif in a 50 nt window (vertical axis) at the indicated position of *AGRN* (horizontal axis) is shown. (**Bottom panel**) Alignment of genomic segments of 7 representative vertebrates to human *AGRN* exons 30 to 37, which were extracted from “Multiple alignments of 100 vertebrate species” in the UCSC genome browser. Pairwise alignments of each species to the human genome (hg38) are shown as a gray−scale density plot, with darker values indicating higher alignment quality. (**B**) RT−PCR showing alternative splicing of endogenous Y and Z exons in SH−SY5Y cells treated with indicated siRNAs. Immunoblotting (**bottom panel**) confirmed the downregulation of PTBP1 and PTBP2. (**C**–**F**) Quantification of (**B**). Mean and SD (*n* = 3 independent experiments) of %Y inclusion and %Z inclusion are indicated. *p*−values of one−way ANOVA (**C**) and two−way ANOVA (**D**–**F**) are indicated above the graph. * *p* < 0.05, ** *p* < 0.01, *** *p* < 0.001, and **** *p* < 0.0001 using Tukey’s post hoc test compared to si−control when applicable.

**Figure 3 ijms-24-07420-f003:**
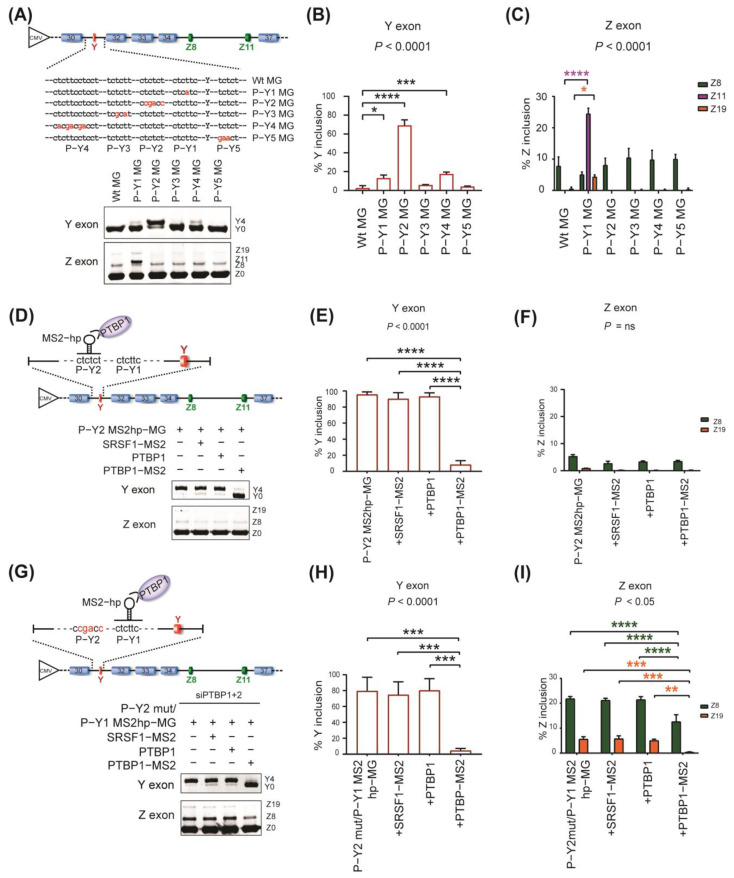
Roles of PTBP1 binding sites around the Y exon for the splicing repression of Y and Z exons. (**A**) Schematic showing wild−type minigene (Wt MG) and mutant minigenes (P−Y1 to P−Y5 MG) carrying mutations at the predicted PTBP1−binding sites around the Y exon. Mutations are indicated in red. RT−PCR (**bottom panel**) showing splicing of the Y exon and Z exons for wild−type and mutant minigenes in SH−SY5Y cells. (**B**,**C**) Quantification of (**A**). Mean and SD (*n* = 3 independent experiments) of %Y inclusion (**B**) and %Z inclusion (**C**) are indicated. (**D**) Schematic (top) showing artificial tethering of fused PTBP1−MS2 coat protein (PTBP1−MS2) to a reporter minigene (P−Y2 MS2hp−MG) carrying an MS2 hairpin loop (MS2hp) at P−Y2. The MS2 hairpin loop is indicated by a stem−and−loop, and the MS2 coat protein is indicated by an inverted U shape. RT−PCR (**bottom panel**) showing splicing of Y exon and Z exons of P−Y2−MS2hp−MG in SH−SY5Y cells co−transfected with each effector. SRSF1−MS2 is a control protein expressing SRSF1 fused with MS2 coat protein. PTBP1 is also a control protein that carries no MS2 coat protein. (**E**,**F**) Quantification of (**D**). (**G**) Schematic (top) showing the reporter minigene, P−Y2mut/P−Y1 MS2hp−MG, carrying an MS2 hairpin loop (MS2hp) at P−Y1. To minimize the effects of PTBP1 binding to other binding sites on the splicing, endogenous *PTBP1*/*2* mRNAs were knocked down using the specific siRNAs and the P−Y2 site was disrupted. RT−PCR (**bottom panel**) showing splicing of the Y exon and Z exons of P−Y2mut/P−Y1 MS2hp−MG in SH−SY5Y cells co−transfected with each effector. (**H**,**I**) Quantification of (**G**). Mean and SD (*n* = 3 independent experiments) are shown in the graphs. *P*−values of one−way ANOVA (**B**,**E**,**H**) and two−way ANOVA (**C**,**F**,**I**) are indicated above the graph. * *p* < 0.05, ** *p* < 0.01, *** *p* < 0.001, and **** *p* < 0.0001 using Tukey’s post hoc test when applicable.

**Figure 4 ijms-24-07420-f004:**
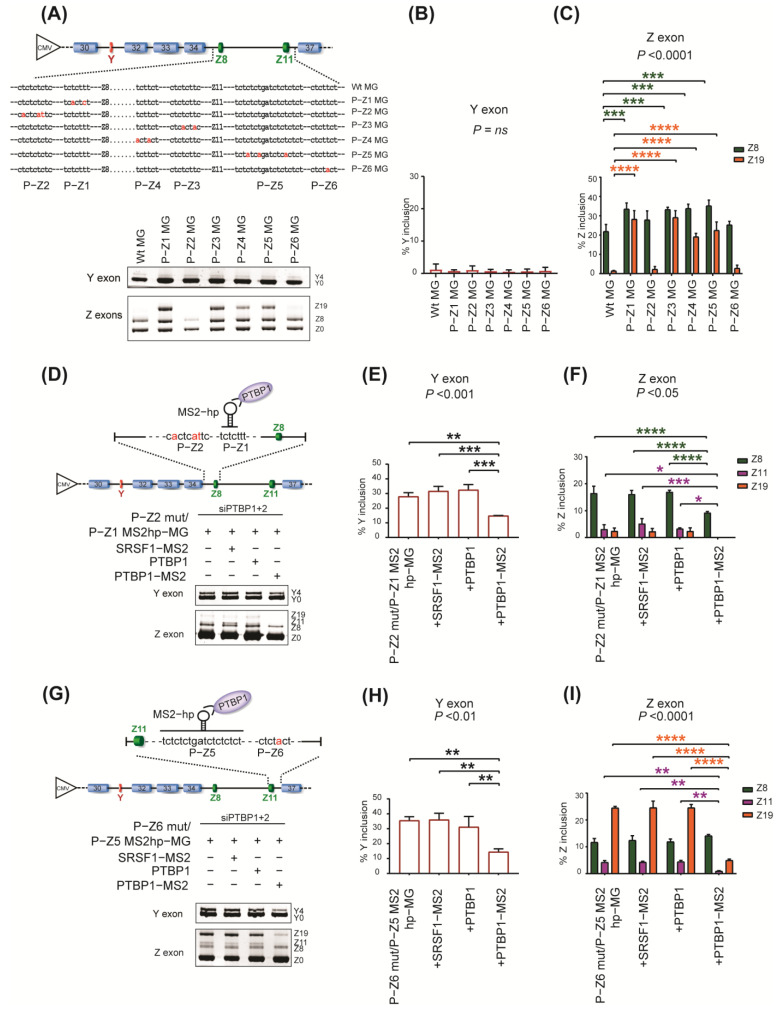
The roles of PTBP1 binding sites around Z exons for the splicing repression of Y and Z exons. (**A**) Schematic showing wild−type minigene (Wt MG) and mutant minigenes (P−Z1 to P−Z6 MG) carrying mutations at the predicted PTBP1−binding sites around Z exons. Mutations are indicated in red. RT−PCR (**bottom panel**) showing splicing of Y and Z exons for wild−type and mutant minigenes in SH−SY5Y cells. (**B**,**C**) Quantification of (**A**). (**D**) Schematic (top) showing the reporter minigene, P−Z2mut/P−Z1 MS2hp−MG, carrying an MS2 hairpin loop (MS2hp) at P−Z1. Cells were co−transfected with the reporter minigene and indicated effectors upon *PTBP1* knockdown, as in Figure 3G. RT−PCR (below) showing splicing of Y and Z exons. (**E**,**F**) Quantification of (**D**). (**G**) Schematic (top) showing the reporter minigene, P−Z6mut/P−Z5 MS2hp−MG, carrying an MS2 hairpin loop (MS2hp) at P−Z5. Cells were co−transfected with the reporter minigene and indicated effectors upon *PTBP1* knockdown. RT−PCR (below) showing splicing of Y and Z exons. (**H**,**I**) Quantification of (**G**). Mean and SD (*n* = 3 independent experiments) are shown in the graphs. *P*−values of one−way ANOVA (**B**,**E**,**H**) and two−way ANOVA (**C**,**F**,**I**) are indicated above the graph. * *p* < 0.05, ** *p* < 0.01, *** *p* < 0.001, and **** *p* < 0.0001 using Tukey’s post hoc test when applicable.

**Figure 5 ijms-24-07420-f005:**
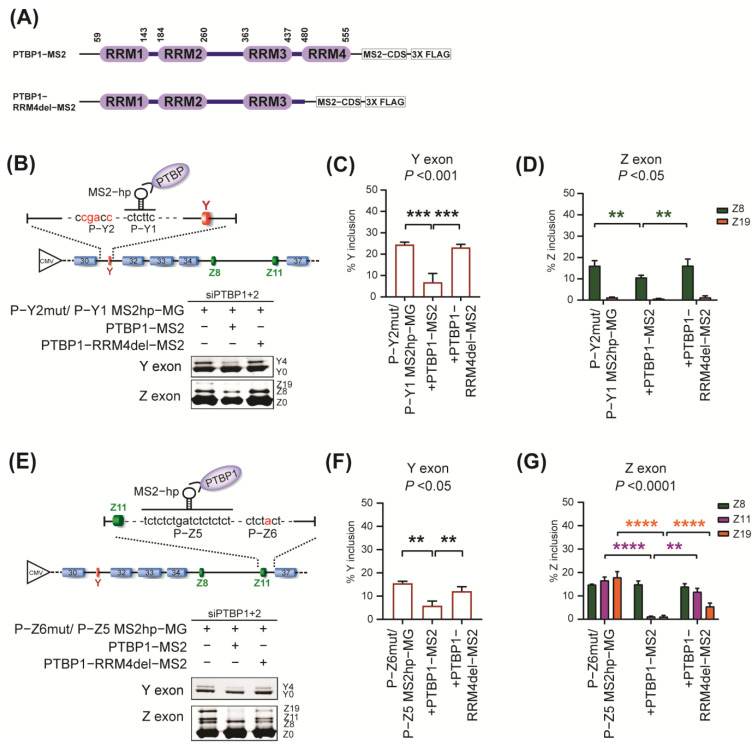
The essential role of RRM4 domain in PTBP1 for the repression of both Y and Z exons. (**A**) Schematic showing full−length PTBP1 (PTBPI−MS2) and RRM4−deleted PTBP1 (PTBP1−RRM4del−MS2). Both are fused with MS2 coat protein and 3X FLAG tag. (**B**) The effect of RRM4 deletion on the P−Y1−mediated splicing repression. PTBP1−MS2 or PTBP1−RRM4del−MS2 is tethered to the P−Y1 site in P−Y2mut/P−Y1 MS2hp−MG upon *PTBP*s knockdown. RT−PCR (below) showing splicing of Y and Z exons. (**C**,**D**) Quantification of (**B**). (**E**) The effect of RRM4 deletion on the P−Z5−mediated splicing repression. PTBP1−MS2 or PTBP1−RRM4del−MS2 are tethered at the P−Z5 site in P−Z6mut/P−Z5 MS2hp−MG upon *PTBP*s knockdown. RT−PCR (below) showing splicing of Y and Z exons. (**F**,**G**) Quantification of (**E**). Mean and SD (*n* = 3 independent experiments) are indicated in the graphs. *P*−values of one−way ANOVA (**C,F**) and two−way ANOVA (**D**,**G**) are indicated above the graph. ** *p* < 0.01, *** *p* < 0.001, and **** *p* < 0.0001 using Tukey’s post hoc test when applicable.

**Figure 6 ijms-24-07420-f006:**
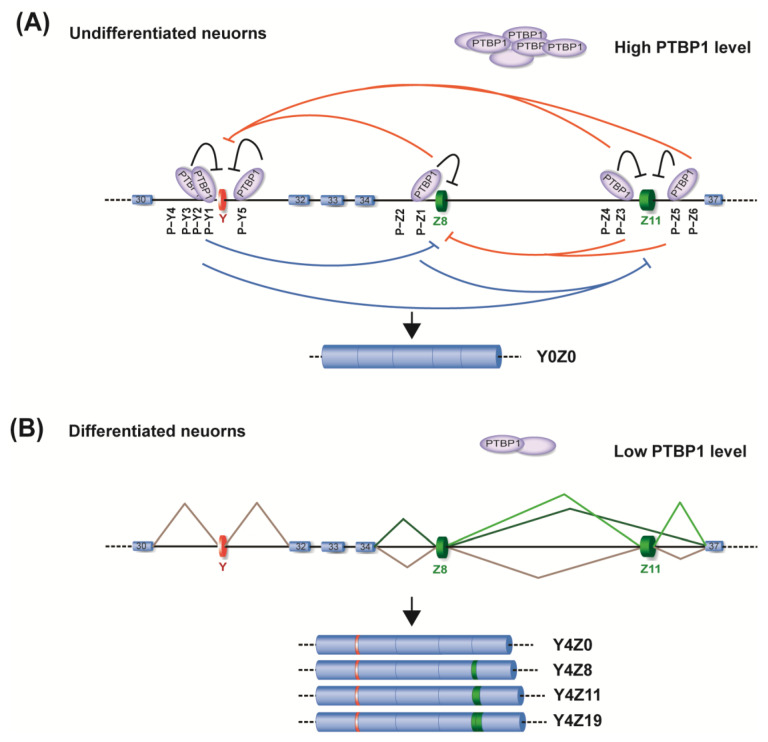
A model for the PTBP1–RNA interaction network that represses the coordinated splicing of *AGRN* Y and Z exons. (**A**) PTBP1 proteins interact with multiple binding sites around Y and Z exons. Each site mediates the repressive activity of PTBP1 against a proximal exon (black lines) as well as distal exons (PTBP1−dependent repression of upstream and downstream exons from distal *cis*−elements is indicated by blue and orange lines, respectively). Multiple bindings of PTBP1 around these exons form a complex network to repress the coordinated splicing of Y and Z exons in a redundant and compensatory manner. (**B**) In differentiated neurons PTBP1 is downregulated, which enhances splicing of Y and Z exons to generate neuronal isoforms of agrin in cooperation with other splicing factors.

## Data Availability

Raw data are available upon request.

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
