# Peer review of "Neural Isoforms of Agrin Are Generated by Reduced PTBP1−RNA Interaction Network Spanning the Neuron−Specific Splicing Regions in AGRN"

_ijms, 2023, doi:10.3390/ijms24087420_

Round 1

Reviewer 1 Report

This manuscript follows the trend of published works from the Ohno laboratory over the past decade. Briefly, this is an in-depth study of the genes, molecular mechanisms and transcripts involved in neuronal splicing patterns of the Agrin gene. The work was carefully performed, the presented gels and sequencing data are clear and coherent and the proposed mechanism of action convincing, which are all positive aspects of the presented work. However, there are weak points as well.

First, it remains unclear if the observed regulatory processes are unique to cholinergic neurons and if they occur in both male and female-originated nerve cells.

Second, it would be interesting to learn if non-neuronal cells with cholinergic features express parallel regulatory processes;

and third, and most importantly, one wonders if altering this splicing pattern can serve as a therapeutic paradigm for congenital myasthenia patients, and if so- how that can be achieved. The authors may wish to refer to these issues in their re-submitted article.

Reviewer 2 Report

Dear Authors,

The manuscript described a detailed alternative splicing of AGRN gene by using neuronal culture model and mini-gene assay. These data are robust and accurate, and the manuscript is well-written. In addition, this model would be a novel evidence such that the activity of PTBP1 dependent splicing repression affects both the proximal and distal exons mediating the intronic cis-elements, even though three constitutive exons are flanking by alternative exons. My only concerns regard this remote control of alternative exon. Is this manuscript a first biochemical evidence that cis-element of RBP binding can control remote exons? Is this a specific role of PTBP1other than the other RBPs? Or is this due to the relatively small of the intron in AGRN gene? Please be discussed about this issue in the discussion. Anyway, it will become appropriate for publication in the International Journal of Molecular Science.

Reviewer 3 Report

Bushra and colleagues investigate the role of the RNA binding protein PTBP1 in regulating the alternative splicing of AGRN pre-mRNA during neuronal differentiation. Agrin function is essential for proper signaling at neuromuscular junctions. Herein, the authors identified conserved cis-elements around Y and Z exons of the human AGRN gene, suitable for PTBP1 binding. Silencing and overexpression experiments, coupled with mutagenesis of the identified consensus motifs, confirmed the role of PTBP1 in the regulation of AGRN splicing. The authors also find that the RRM4 domain of PTBP1 is essential for its role as splicing repressor, as previously shown by other laboratories.  The authors conclude that physiological downregulation of PTBP1 during neuronal differentiation is essential to promote the coordinated inclusion of Y and Z exons for the generation of neuron-specific agrin isoforms.

Overall, the study is well-designed and the experiments well-conducted. Thus, I do not have major concerns.

I would only suggest implementing discussion with structural insights on the role of the RRM3 and RRM4 of PTBP1 in splicing repression. The relevance of PTBP1 mediated regulation of AGRN splicing for neuromuscular junction-associated diseases could also be discussed.

Lastly, I would suggest simplifying and ameliorating Figure 6, for the sake of comprehension.

Round 2

Reviewer 1 Report

the authors did make an effort to adhere to the review comments, which is appreciated.